# Molecular Mechanisms of “Antiphospholipid Antibodies” and Their Paradoxical Role in the Pathogenesis of “Seronegative APS”

**DOI:** 10.3390/ijms21218411

**Published:** 2020-11-09

**Authors:** Roberta Misasi, Agostina Longo, Serena Recalchi, Daniela Caissutti, Gloria Riitano, Valeria Manganelli, Tina Garofalo, Maurizio Sorice, Antonella Capozzi

**Affiliations:** Department of Experimental Medicine, University of Rome “La Sapienza”, 00161 Rome, Italy; agostina.longo@uniroma1.it (A.L.); serena.recalchi@uniroma1.it (S.R.); caissutti.1704052@studenti.uniroma1.it (D.C.); gloria.riitano@uniroma1.it (G.R.); valeria.manganelli@uniroma1.it (V.M.); tina.garofalo@uniroma1.it (T.G.); maurizio.sorice@uniroma1.it (M.S.); antonella.capozzi@uniroma1.it (A.C.)

**Keywords:** SN-APS, β2GPI, PTMs, “non-criteria” aPL, lipid rafts, TLR-4 pathways

## Abstract

Antiphospholipid Syndrome (APS) is an autoimmune disease characterized by arterial and/or venous thrombosis and/or pregnancy morbidity, associated with circulating antiphospholipid antibodies (aPL). In some cases, patients with a clinical profile indicative of APS (thrombosis, recurrent miscarriages or fetal loss), who are persistently negative for conventional laboratory diagnostic criteria, are classified as “seronegative” APS patients (SN-APS). Several findings suggest that aPL, which target phospholipids and/or phospholipid binding proteins, mainly β-glycoprotein I (β-GPI), may contribute to thrombotic diathesis by interfering with hemostasis. Despite the strong association between aPL and thrombosis, the exact pathogenic mechanisms underlying thrombotic events and pregnancy morbidity in APS have not yet been fully elucidated and multiple mechanisms may be involved. Furthermore, in many SN-APS patients, it is possible to demonstrate the presence of unconventional aPL (“non-criteria” aPL) or to detect aPL with alternative laboratory methods. These findings allowed the scientists to study the pathogenic mechanism of SN-APS. This review is focused on the evidence showing that these antibodies may play a functional role in the signal transduction pathway(s) leading to thrombosis and pregnancy morbidity in SN-APS. A better comprehension of the molecular mechanisms triggered by aPL may drive development of potential therapeutic strategies in APS patients.

## 1. Introduction

APS is an autoimmune disease, characterized by recurrent thrombosis and/or obstetrical morbidity and a series of systemic manifestations induced by the persistent presence of aPL, including lupus anticoagulant (LA), anti-β2-GPI and/or anti-cardiolipin (aCL) antibodies. 

In 2006, shared criteria were defined for the diagnosis of APS [1]. Clinical criteria include venous or arterial thrombosis and complications of pregnancy, including loss of pregnancy or premature birth. Laboratory criteria include at least one positive for aPL. The aPL recognized in the international criteria include antibodies (IgG or IgM) aCL higher than 40 phospholipid units IgG or IgM or anti-β2-glycoprotein I (β2-GPI) antibodies with titers above the 99th percentile and lupus LA detected according to guidelines published by the International Society on Thrombosis and Hemostasis (ISTH). Since persistent positivity is required for diagnosis, these tests should be present on two or more occasions at least 12 weeks apart.

Classification criteria for diagnosis of APS require that a patient has the combination of at least one laboratory and one clinical (arterial or venous thrombosis and/or pregnancy morbidity) criteria, as reported by the revised international classification criteria for APS [1,2,3]. However, the clinical spectrum of the disease may include additional manifestations which may affect various organs and cannot be explained exclusively by a prothrombotic state [4].

In the last decade, the significance of aPL persistency and accumulation (the co-presence of aPL criteria: aCL, anti-β2-GPI of the IgG or IgM subtypes and circulating LAC) was evaluated, especially regarding the risk of APS evolvement, although their functional role for assessment of the specific APS-related manifestations is not always clear. However, “seropositivity” of all the three-classification criteria, termed the “triple positive”-variant, correlates with a more aggressive disease. It requires specific therapeutic interventions, such as anti-coagulant drugs [5,6]. Several studies revealed that aPL are a heterogeneous group of autoantibodies that have a clear association with thrombosis and pregnancy morbidity and are directed against proteins, complexes of phospholipids and phospholipid-binding proteins or phospholipids [7,8]. Proteins as a target of “aPL antibodies” are so far identified not only as β2-GPI [9], but also as annexin A5 [10], annexin A2 [11], prothrombin (PT) [12], protein C [13] and protein S [14]; as complexes of phospholipids and phospholipid-binding proteins, prothrombin/phosphatidylserine (PT/PS) [15] and vimentin/cardiolipin [16]; as phospholipids, in addition to cardiolipin, phosphatidylserine (PS) [7], phosphatidylethanolamine (PE) [17] and lysobisphosphatidic acid (LBPA) [18]. 

However, in daily clinical practice, it is possible to identify patients with clinical symptoms suggestive for APS, but persistently negative for classical laboratory criteria. This patients’ population was referred to as seronegative SN-APS, although “new” aPL specificities have been described in these patients. In fact, in 2003 it was deemed necessary to classify a new entity of the syndrome (SN-APS) that it is still the subject of debate. G. Hughes, M. Khamashta and other research teams speculated that this clinical entity could be explained by the presence of non-criteria aPL, which may not have been considered in the serological battery of the tests [19,20].

Despite the strong association between aPL and thrombosis, the exact pathogenic mechanisms underlying thrombotic events and pregnancy morbidity in the course of APS have not been yet fully elucidated and more than one mechanism may be involved, such as the exposure to some environmental agents, such as infections, in susceptible individuals [21,22,23]. 

Antiphospholipid antibodies binding β2-GPI, may contribute to thrombotic diathesis by interfering with hemostasis [24,25]. Numerous studies highlight the role of activation of monocytes, endothelial cells, platelets and/or complement, as well as the induction of a prothrombotic state caused by interference with coagulation cascade proteins. aPL interact with endothelial cells inducing adhesion molecules, such as intercellular cell adhesion molecule-1 (ICAM-1), vascular cell adhesion molecule-1 (VCAM-1) and E-selectin expression and proinflammatory cytokines release, thus inducing a proinflammatory and procoagulant cell phenotype [22]. Moreover, aPL may activate platelets, with an increase of glycoprotein IIb–IIIa expression [23], thromboxane A2 synthesis [26] and platelet factor-4 secretion, a chemokine with procoagulant and prothrombotic effects [27,28]. Several reports show that in monocytes and endothelial cells, anti-β2-GPI antibodies may lead to an up-regulation of Tissue Factor (TF), which plays a pivotal role in initiating the extrinsic coagulation cascade [29].

Another mechanism involves the Annexin A5 protein, this is a natural and physiological anti-coagulant which binds to PS on the cell surface, and forms a shield to prevent the activation of procoagulant complexes. Anti-β2-GPI/β2-GPI complex can disrupt this anticoagulant shield, exposing procoagulant PS, hence predisposing to thrombosis [30,31]. 

To explain the pathogenesis of thrombosis in APS, a model was suggested: “the first hit and second hit”. According to this model, aPL (the “first hit”) destroy the integrity of the endothelium inducing a procoagulant phenotype, nevertheless, thrombosis takes place only in the presence of an initiating factor (the “second hit”), as a consequence of smoking, infection, oxidative stress or inflammation [32]. Thus, in some cases, aPL cannot be considered pathogenic in those patients who are termed as “asymptomatic carriers” [33]. One of the molecular mechanism involving endothelial cells and compatible with this pathogenetic hypothesis is represented by the activation and signaling through Toll-like Receptor 4 (TLR-4) that drives ultimately to an excessive release of proinflammatory cytokines (“cytokine storm”) and an increase in the production of procoagulant factors, as well as the expression of cell-adhesion molecules [34,35].

Moreover, a variant defined catastrophic APS (CAPS) has been described. It is characterized by the development of excessive thrombosis at multiple sites, usually affecting small vessels and leading to multi-organ dysfunction and organ failure. The classification criteria for CAPS include the involvement of three or more organs, systems and/or tissues, the development of multiple manifestations simultaneously or in less than one week, histopathological confirmation of microvascular thrombosis and the presence of aPL. The high rate of mortality and the sporadic distribution of cases have made it difficult to study the physiopathology of this condition. In 1998 Kitchens postulated the theory of thrombotic storm, according to which existing vascular occlusions promote an anti-fibrinolytic state through an increase in plasminogen activator inhibitor and a consumption of procoagulant factors, creating an imbalance in the homeostasis of the coagulation system, favoring further thrombus formation [36].

Undoubtedly, since then, the study of the pathogenetic mechanisms of the syndrome has certainly benefited from the diagnostic hypothesis of the non-criteria aPL. In this way, it has been possible to reconstruct several pathogenetic pathways that lead to thrombogenesis or obstetric complications, directly or indirectly connected to aPL. For example, the signaling cascade target of rapamycin (mTOR) in mammals, has added much to our understanding of the pathophysiology of the disease and has encouraged researchers to study the treatment options [37].

The finding of new autoantibodies to use as new markers and scoring tools is always very interesting, especially by investigating their role in the pathogenetic mechanisms leading to thrombosis dysfunction and pregnancy morbidity in SN-APS patients.

## 2. β2-GPI Conformations in the Pathogenesis of APS

If we consider the proposed prothrombotic mechanism of “two hits”, the existence of aPL (criteria and non-criteria) remains the main contributing factor and the “first hit” in the pathogenesis of APS [32].

The pathogenetic mechanism of aPL and mostly anti-β2-GPI antibodies includes, among other mechanisms, the alteration of both phases of the coagulation cascade: the fluid phase, by interfering with the Vitamin K dependent protein C and annexin A5, and the cellular phase, by activating platelets, inducing the expression of TF on monocytes and endothelial cells with consequent activation of complement [4,24]. 

β2-GPI, also known as apolipoprotein H, is a 45 kDa plasma glycoprotein with affinity for anionic phospholipids. It is mainly synthetized in the liver and has a plasma concentration of approximately 200 μg/mL. It has been suggested that β2-GPI may inhibit platelet prothrombinase activity [38] and ADP-mediated platelet aggregation [39]. Moreover, because of its high affinity for anionic phospholipids, it was thought that β2-GPI, by inhibition of the contact phase activation of coagulation, could play a role in maintaining the hemostatic balance [40]. It has also been suggested that β2-GPI may function as a scavenger protein. Indeed, the molecule was shown to react with lipopolysaccharide (LPS), facilitating its phagocytosis by monocytes-macrophages [41]. Consistent with such scavenging activity is also the binding of β2-GPI to apoptotic material. During apoptosis, the reorganization of membrane lipid bilayer determines the exposure of PS, which is generally distributed in the cytoplasmic leaflet of the plasma membrane of cells, on the outer cell surface. β2-GPI binds to PS-exposing vesicles or apoptotic cells, promoting their engulfment by phagocytes [42]. 

Although β2-GPI was first described in 1961 by Schultze et al. [43], the interest in this protein increased significantly only about thirty years after its discovery, when it was recognized as the major antigenic target in APS [44,45,46]. β2-GPI consists of a single polypeptide chain of 326 amino acid residues and comprises five domains, arranged like a pearl necklace. The first four domains are short consensus repeats from the complement control protein (CCP) module type, also known as “sushi domains”. CCPs are modules of about 60 amino acids common in many proteins involved in the regulation of complement activation. On the contrary, the amino acid sequence of the fifth domain (DV) gives it features different from the other domains. The DV, in fact, is stabilized by three internal disulfide bonds instead of two. It carries a definite positive charge and two portions, located at the lower part of the DV, constitute an excellent counterpart for interactions with negatively charged amphiphilic substances. Therefore, the DV is responsible for the binding of β2-GPI to anionic phospholipids on cell membranes. β2-GPI exists in two interconvertible biochemical variants, oxidized (54%) and reduced (46%), depending on the integrity of the disulfide bonds [47,48]. In the oxidized form, 11 disulfide bonds are formed. In the reduced form, the disulfide bonds C288–C326 in DV and C32–C60 in DI are individually or simultaneously broken. After the acknowledgment of the central importance of β2-GPI in APS, many studies followed to identify the portions of the protein involved in the binding of anti-β2-GPI antibodies. It has been shown that antibodies can be directed against epitopes located in different domains of the protein. However, robust evidence indicates that the immunodominant epitope is in the first domain (DI) of the protein. Using domain-deletion mutants, Iverson et al. showed for the first time that anti-β2-GPI antibodies recognize an epitope on DI [49]. To further explore the fine specificity of these anti-DI antibodies, Iverson et al. constructed several other mutants with point mutations in domain I [50]. The main epitope was reported to involve the amino acid residues R39-R43 and D8-D9, as well as the linker region between domains I and II.

Some relevant studies were undertaken to evaluate the clinical relevance of domain specificity profiling of anti-β2-GPI IgG antibodies in APS. Of particular interest are the anti-β2-GPI domain I antibodies (anti-β2-GPI DI), in which significant pathogenetic correlations between thrombosis and fetal loss have been documented, in particular with antibodies directed against domain I and not with other β2-GPI domains [50,51,52].

Multiple studies have led to the identification of anti-β2-GPI DI antibodies as the key pathogenic subset of autoantibodies in APS. In vitro, anti-β2-GPI DI IgG were found to have lupus LAC activity and strongly associate with thrombosis. Moreover, recombinant human β2-GPI DI was found to lower aPL-induced thrombosis [53]. Furthermore, anti-β2-GPI DI IgG have been shown to confer increased resistance to the anticoagulant properties of both annexin A5 [54] and activated protein C [55]. Anti-β2-GPI antibodies displaying LAC activity were also demonstrated to abrogate the β2-GPI-mediated inhibitory effect on von Willebrand factor dependent platelet adhesion and aggregation [56]. On the other hand, they may also interfere with the intrinsic anti-thrombotic functions of β2-GPI. Importantly, both affinity-purified anti-β2GPI DI IgG from APS serum and human monoclonal anti-β2GPI DI IgG have been shown to induce thrombosis and/or fetal loss in mice APS models of venous thrombosis [57,58]. Agostinis et al. proved that a human monoclonal antibody directed against β2-GPI DI exhibits complement-dependent procoagulant and pro-abortive effects, and a variant of this antibody, lacking the CH2 domain, is effective in preventing blood clot formation and fetal loss induced by aPL [59]. Therefore, it is now widely accepted that autoantibodies directed against β2-GPI DI can drive APS pathogenesis and are associated with thromboembolic events. In a large cohort study, Andreoli et al. have demonstrated that IgG targeting β2-GPI DI represent the prevalent subset not only among APS patients but also among individuals with autoimmune conditions with any clinical sign indicative of APS. They proposed that the ratio between antibodies to DI and those targeting DIV/V of β2-GPI may be useful in determining the pathogenic potential of anti-β2-GPI antibodies and in discriminating between autoimmune disorders and non-immune conditions [60]. Although it is widely recognized that the first domain of β2-GPI plays a fundamental role in the pathogenesis of the syndrome, multiple studies suggest that aPL directed against different domains of the protein may play a pathogenic role [61]. In particular, Murthy et al. demonstrated that IgA anti-β2-GPI directed to domain IV/V of the molecule represent an important subgroup of clinically relevant aPL, which may play a pathogenic role, as revealed in a mouse model of thrombosis [62]. Other researchers have recently shown that IgA anti-β2-GPI antibodies found in APS patients with clinical signs of thrombosis bind to three sites in D3, D4 and D5 [63]. Moreover, anti-β2GPI (ILA-1, ILA-3 and H-3 MoAb), that are able to interact specifically with three different hexapeptides corresponding to distinct epitopes located in domains I-II, III and IV of the molecule [64,65], are able to activate endothelial cells in vitro and induce experimental APS by passive transfer [66]. 

The antigenic epitopes of β2-GPI domains have been defined as cryptic, since they are exposed on the outer surface of the protein only when the latter is in the open conformation. On the contrary, these epitopes are either buried by domain V in the circular form or shielded by the *N*-linked glycosylations in the S-twisted form. Indeed, β2-GPI can adopt multiple conformations (J-elongated, S-twisted and O-circular), resulting in different exposures of each of its domains to the solvent [67,68]. The crystal structure of β2-GPI revealed a hockey stick-like shape of the molecule, in which the first four domains are stretched along their long axis while the fifth domain is at a right angle to the other ones. This structure is thought to be the immunogenic conformation of β2-GPI that interacts with anti-β2-GPI antibodies, which forms when the protein binds to the membranes. The O-circular structure of β2-GPI was originally proposed by Koike et al. in 1998 [69] to explain the lack of binding of antibodies to β2-GPI in solution, and then it was captured by electron microscopy by Agar et al. [67]. This closed circular conformation has been shown to result from an intramolecular interaction between DI and DV, which in turn causes the epitope on DI to be hidden by DV. It is believed to be the conformation that the protein takes in the free form in plasma.

All these structural studies have led to the conclusion that in vivo β2-GPI can exist in both closed and open conformations [67], and that the interaction of the molecule with the surrounding microenvironment determines its structure at any given time. Specifically, the commonly accepted model predicts that in plasma β2-GPI is present primarily (>90%) in the closed circular conformation, with the epitope on domain I not accessible for autoantibodies. This assumption is consistent with the observation that circulating immunocomplexes between β2-GPI and antibodies are usually not easily detected in APS patients’ sera. In contrast, when β2-GPI interacts with negatively charged surfaces, such as anionic phospholipids, a conformational change occurs and β2-GPI adopts an open elongated form; the epitope on domain I is exposed to the solvent and antibodies can recognize it and bind to β2-GPI (Figure 1).

In a recent work, Ruben et al. have studied the structural architecture of β2-GPI in solution, under conditions relevant to physiology. Their results were unexpected and not consistent with what is widely accepted in this field. Indeed, they found that in solution, the monomeric oxidized form of β2-GPI, which accounts for 54% of the protein in human plasma, adopts a J-elongated conformation under physiological pH and salt concentrations, neither O-circular nor S-twisted. The alternative model proposed that the open form of oxidized β2-GPI pre-exists and predominates in human plasma, thus suggesting that the opening of the protein structure and the relocation of carbohydrate chains, away from domain I upon β2-GPI binding to membranes, are neither necessary nor sufficient so that the protein exposes the cryptic epitope [70]. 

All the studies concerning the structure of the β2-GPI and its different conformations lead to evidence biochemical differences between the physiological and pathogenic protein. Several studies have suggested a pathogenic role for post-translational modifications (PTMs) of proteins as modulators of protein structure and function. In particular, this mechanism may be a highly specific phenomenon responsible for pathogenic structure of β2-GPI making it more antigenic.

## 3. Post-Translational Modifications and Pathogenesis of APS

Post-translational modifications of proteins, defined as covalent modifications of self-proteins, take place at specific amino acids and occur in many physiological conditions representing a potential trigger of several autoimmune diseases. Many of the notable modified self-proteins triggering autoreactive B and T cell responses, in many cases represent specific diagnostic biomarkers as well as a reflection of the disease. Some PTMs can be directly affected by tissue reactive oxygen species (ROS) and/or by inflammatory microenvironments (carbonylation, methylation, isoaspartylation, deamidation), or be influenced by more indirect downstream pathways affected by ROS (acetylation, glycosylation, phosphorylation, citrullination) [71,72]. Many PTMs of self-proteins are involved in APS, as in vivo triggers of autoimmune reactions, and the modified proteins can play a role in the pathogenesis of APS, contributing to design more efficient diagnostic/prognostic tools and more targeted therapeutic approaches. 

The known PTMs in APS mainly involve β2-GPI; in particular, the modification via thiol-exchange reactions is an important event in the setting of APS thrombosis. In addition, also other PTMs, including sialylation, acetylation and the addition or loss of carbohydrate chains affects β2-GPI conformation and immunoreactivity. Interestingly, some PTMs of the protein result in adoption of an open configuration that may expose the cryptic epitope and facilitate autoantibody binding [73].

Ioannou et al. showed that the relative amount of oxidized β2-GPI is increased in APS patients with a thrombotic history compared to healthy volunteers [74]. In previous studies, we demonstrated that oxidized β2-GPI induces dendritic cell maturation and promotes Th1 polarization with IL-1β, IL-6, IL-8, IL-12 and TNF-α release [75]. Moreover, a partial or complete removal of the carbohydrate chains (deglycosylation) or reducing number of sialic acids in glycan structure at Asn143 (desialylation) may lead to conformation instability of β2-GPI as well as the process of glycation. Previously, the presence of several potential glycation sites within the primary structure of β2-GPI was demonstrated [76]. Interestingly, glycated β2-GPI induced phenotypical and functional maturation of dendritic cells, involving the activation of p38 mitogen-activated protein kinase (MAPK), ERK and nuclear factor-kappa B (NF-κB). It also induced a significant up-regulation of RAGE, the receptor for advanced glycation end products. These in vitro data suggested that conformational changes induced by glycation confer to β2-GPI the ability to act as an inflammatory stimulus able to activate immunogenic dendritic cells and lead to T-cell response [77]. In fact, antibodies directed to glycated β2-GPI were detected in patients with APS, suggesting that glycated-β2-GPI is a target antigen of humoral immune response in the syndrome. A functional role for glycated-β2-GPI is evident from the significant correlation found between anti-glycated β2-GPI antibodies and several clinical manifestations of APS, including venous thrombosis and seizure [76]. 

Moreover, lysine residue acetylation, as well as biotinylation of β2-GPI and in general protein misfolding, could alters the β2-GPI conformation with potential implications for the signal transduction pathway triggered by the protein [73]. 

In addition to β2-GPI, PTMs of other self-proteins, such as vimentin and annexins, may play a role in the immune response during APS. The presence of autoantibodies directed against the vimentin/cardiolipin complex has been also demonstrated in a high percentage of APS patients, in association with both arterial and venous thrombosis [16]. Vimentin is a dynamic protein and it was described its role in the pathogenesis of inflammation and several autoimmune diseases. Different studies showed that PTMs of the protein are important for the induction of specific antibodies in various diseases such as Rheumatoid Arthritis, Systemic Lupus Erithematosus (SLE), APS and others [78].

The main PTMs found for vimentin are phosphorylation, sumoylation, citrullination, carbamylation and acetylation [78]. In particular, citrullination, carbamylation and acetylation, as well as the binding to phospholipids are involved in conformation changing of vimentin, promoting the exposure of neoepitopes to the immune system [79]. In fact, Alessandri et al. found antibodies directed to mutated citrullinated vimentin in 26.6% of patients with APS with a strong association with arthritis. These findings indicated a role for citrullination in the pathogenesis of APS and in the development of joint involvement [80]. Indeed, citrullination and/or other PTMs of intracellular proteins, including vimentin, would explain their expression on the plasma membrane where can bind anionic phospholipids. 

Annexins are a group of 12 highly conserved proteins characterized by their ability to bind phospholipids, predominantly in a calcium-dependent manner. Anti-annexin antibodies have been found in patients with APS [81]. A previous study hypothesized that annexin A1 undergoes PTM as a part of neutrophil extracellular traps (NETs) that are produced in response to viral, bacterial and/or inflammatory triggers [82] and in particular, the focus is on the process of citrullination of annexin A1, which takes place within NETs. 

Post-translationally modified proteins can break tolerance and induce the autoimmune spiral. To better understand the pathogenesis of APS, it would be interesting to study modified/misfolded proteins and to analyze the mechanisms causing their accumulation, such as defective protein catabolism or protein transport. In addition, NETosis is considered one of the main cellular processes to be evaluated as a source of modified antigens. This is a compelling argument for a role of NETs in autoimmunity, since posttranslational modified proteins presented in an inflammatory environment may become antigenic and, exposed to professional cells leading to a specific immune response and autoantibodies production [73].

## 4. Signal Transduction Pathways Triggered by aPL

Only in the last 15 years the signal transduction pathway(s) triggered by aPL have been studied in detail, allowing to better clarify the functional role of aPL and, consequently, the pathogenic mechanisms of APS. Raschi et al. first demonstrated that aPL binding to endothelial cells (ECs) activates the TLR-4 transduction signaling pathway, as revealed by transiently co-transfecting microvascular endothelial cells (HMEC-1) with dominant-negative constructs of different components of the pathway (ΔTRAF2, ΔTRAF6, ΔMyD88) [35]. In particular, they demonstrated the activation of the myeloid differentiation factor 88 (MyD88), phosphorylation of interleukin-1 receptor–associated kinase (IRAK), the first kinase specifically recruited by receptors of the IL-1/TLR superfamily and consequent translocation of NF-κB. Several receptors have been suggested to mediate β2-GPI/ECs binding, including mainly Annexin A2 and LRP8 [83]. AnnexinA2 provides a high-affinity binding site for β2-GPI but, since it does not span the cell membrane, an adaptor protein is required to trigger signal. Consistent evidence supports the role of TLR-4 as co-receptor for β2-GPI on ECs [84]. Engagement of TLR-4 activates MyD88-dependent and MyD88-independent pathways, which lead to the activation of downstream targets, such as MAP kinases. 

These findings were confirmed and extended by further studies which demonstrated that human anti-β2-GPI antibodies are able to induce proinflammatory and procoagulant monocytic phenotype, characterized by the release of TNF-α and TF [85]. Indeed, these antibodies are able to trigger signal transduction pathway(s) in monocytic cells. In this regard, the first indication derived from the observation that anti-β2-GPI binding was revealed to occur within lipid rafts, specialized portions of cell plasma membrane implied in signal transduction [86]. Lipid rafts are enriched in cholesterol, gangliosides, ceramides and sphingomyelin. These regions are characterized by a highly ordered and tightly molecular structure as compared to the surrounding bilayer. A large variety of proteins involved in signal transduction pathway(s) has been detected in these microdomains, including tyrosine kinase receptors, mono- (Ras, Rap) or heterotrimeric G proteins, Src-like tyrosine kinases (lck, lyn, fyn), protein kinase C isozymes, GPI-anchored proteins and adhesion molecules. It is now established that lipid rafts can function as recruitment sites for specific proteins, triggering and guiding cell signaling [86]. Biochemical analyses indicated that the only form of β2-GPI within lipid rafts is a dimeric form of the protein [85], which may represent the oxidized protein spontaneously occurring in the culture medium [75]. Thus, β2-GPI interacts with lipid rafts only after dimerization, possibly as a consequence of conformational changes, suggesting that oxidized β2-GPI is able to trigger signal transduction pathways [75,87] and that β2-GPI dimers mimic in vitro the effects of β2-GPI-anti-β2-GPI antibody complexes [87]. In addition, AnnexinA2, which is the main receptor for β2-GPI [88], was found to be highly enriched in lipid raft fractions, where it coimmunoprecipitated with β2-GPI. This finding supported the view that the antibodies recognize β2-GPI that is coupled with its own receptor [35] and that this complex may induce cell signaling, as suggested by Meroni et al. [89]. Interestingly, lipid raft involvement was mandatory for triggering this signaling pathway, since anti-β2-GPI antibodies failed to induce IRAK phosphorylation and NF-κB translocation in the presence of the raft-disrupting agent methyl-beta-cyclodextrin [85]. This finding strongly supported the view that lipid rafts play a key role in the signal transduction pathway triggered by anti-β2-GPI antibodies and that raft-dependent anti-β2-GPI triggering resulted in the release of TNF-α, as well as of TF, which may contribute to the pathogenesis of thrombosis in patients with APS [90,91]. 

Recent evidence showed that anti-β2-GPI antibodies may also trigger a similar signal transduction pathway in human platelets, which involves IRAK phosphorylation and NF-κB activation, followed by TF expression, suggesting that platelets may play a role in the pathogenetic mechanism of APS [92] (Figure 2).

Additional studies demonstrated that aPL may also induce the translocation of TLR-7 or TLR-8 from the endoplasmic reticulum to the endosome [93], these effects depend on the uptake of aPL into the endosome, subsequent activation of endosomal NADPH oxidase and generation of superoxide. A pivotal role for TLR-9 pathway in TF production by monocytes was also recently reported [94].

In the past years, a molecular mimicry has been shown between β2-GPI and bacterial antigens [95] or microbial products. It has been hypothesized that β2-GPI may interact with TLR-4 or that anti-β2-GPI may crosslink the molecule and TLR-4, thus triggering the inflammatory cascade [96]. Indeed, TLRs are important components of innate immunity, recognizing specific microbial products and driving the inflammatory response by interaction with specific ligands [97]. Moreover, it was observed that antibodies to a β2-GPI domain I peptide (PDDLPFST), that shares 88% identity with an epitope within the extracellular domain of TLR-4, were able to engage TLR-4 directly, inducing a proinflammatory phenotype [98]. 

However, additional signaling pathways may be involved in the procoagulant effect of aPL. Indeed, TF expression may be induced in monocytes through the simultaneous activation of NF-κB proteins (via the p38 MAPK pathway) and of the MEK-1/ERK pathway [99]. This may reflect the presence of more than one synergic activating pathway. In this concern, activation of the mTOR pathway plays a role in endothelial proliferation and intimal hyperplasia in anti-PL-positive patients, which leads to multiple potential outcomes, including micro thrombosis. Indeed, aPL, when incubated with vascular endothelial cells, stimulated mTOR through the phosphatidylinositol 3-kinase–Akt pathway [37], leading to cell proliferation. This finding was further supported by the observation that sirolimus, an mTOR complex inhibitor, reduced endothelial cell proliferation and vascular lesions in patients with APS. The role of mTOR pathway in aPL signaling was also confirmed in monocytes in which treatment with anti-β2-GPI/β2-GPI complex induced mTOR activation as well as expression of TF and IL-8. This effect could be attenuated by pretreatment with the mTOR inhibitor rapamycin [100].

However, several events are likely to play a role in pathogenesis, including endothelial, monocyte and platelet cell activation. In addition, recent studies focused on neutrophil release of chromatin in the form of NETs as an additional key contributor, this is a process responsible for PTMs and thus beginning of modified antigens [101]. Thus, the joint contribution of signaling pathways involved in both innate and adaptive immunity may play a role in the crucial steps of APS pathogenesis. 

## 5. Signaling and Pregnancy Complications

Pregnancy complications are hallmarks of the obstetric subset of APS called OAPS. The association of circulating aPL and recurrent miscarriages has been described in several studies [102,103,104], and OAPS is considered as a major acquired risk factor to be evaluated in women with recurrent pregnancy loss [105]. In fact, the knowledge and the deepening of the pathogenetic mechanisms of aPL, involved in the damage and abnormal development of the placenta, was and will be mostly useful for managing of APS in pregnancy. Pregnancy morbidity in APS include unexplained recurrent early pregnancy loss (in the first trimester) and fetal death or premature birth (in the second or third trimesters) with symptoms related to placental insufficiency and dysfunction, such as intrauterine growth restriction (IUGR), preeclampsia, eclampsia or other [2,3].

Pathogenesis studies describe the two-hit hypothesis as the main mechanism underlying manifestations of vascular APS. It seems, however, that the obstetrical signs of the syndrome cannot be explained by this model. In fact, passive infusion of IgG fractions with aPL activity alone has shown to induce fetal loss in naïve pregnant mice without requiring any additional factor (i.e., without the second hit). β2-GPI is widely expressed in placental tissues, even under physiological conditions. Thus, it might be that the high expression of β2-GPI at the placental level, together with the hormonal and vascular modification linked to the pregnancy, is enough to promote the pathogenic activity of the autoantibodies without any supplementary factor [106].

Several researchers described the pathogenic role of anti-β2-GPI antibodies in the morbidity of pregnancy. Indeed, the anti-β2-GPI are able to bind to the monolayers of endothelial or trophoblast cells and, by destroying the anticoagulant shield of Annexin A5, to induce a procoagulant state in the placenta leading to thrombosis and heart attack [107]. In general, the disruption of the balance between pro-inflammatory and anti-inflammatory processes significantly affects the fate of the embryo.

OAPS pathogenesis related to the first trimester pregnancy is mainly based on interaction of aPL with β2-GPI, leading to an alteration of trophoblast function and appears different from late pregnancy morbidity [108,109]. In particular, in early pregnancy loss, a role was described for aPL on placentation and apoptosis of trophoblast cells. In vitro studies demonstrated that aPL binding to β2-GPI, via apolipoprotein E receptor 2 (ApoER2), decreases proliferation and migration of trophoblast cells, promoting antiangiogenic molecule release and destroying trophoblast-endothelial interactions [110,111,112]. Instead, thrombotic mechanisms are prominent in placental dysfunction, cause of late obstetric manifestations, such as IUGR and preeclampsia [108,113]. 

Furthermore, as already pointed out for APS, the inflammatory signaling induced by aPL also plays an important and central role in OAPS, at the level of trophoblast cells and maternal-fetal interface [114]. As previously mentioned, aPL are able to activate TLRs and NLRP3 inflammasome resulting in an increase of inflammatory cytokines and chemokines (IL-1β and IL-8) [105,106]. Moreover, trophoblast function is reduced by aPL that, through ApoER2, lead to reduction of IL-6 promigratory molecule, STAT-3 activity and human chorionic gonadotropin (HCG) production [96,115,116]. Placentae of APS patients are structurally modified with an increase of syncytial knots as damage markers. These are aggregations of nuclei, probably derived from shedding from placenta of large extracellular vesicles, named syncytial nuclear aggregates (SNAs), due to the action of aPL, which can be internalized by syncytiotrophoblast and disrupt mitochondria, causing abnormal cell death [117,118]. Several reports demonstrated a role for aPL in the activation of the complement cascade in placental insufficiency. Others showed an activation of complement component C5 and its cleavage product C5a, together with the recruitment and activation of polymorphonuclear neutrophil (PMN) and monocyte, with a consequent release of ROS, TNF, antiangiogenic factors and TF [119,120,121,122,123]. 

The key role of the complement system in APS pregnancy morbidity has also been studied and demonstrated in murine models, highlighting how mice with genetic complement deficiency or treated with complement blocking substances were protected from pregnancy complications induced by aPL [124,125,126]. 

A further effect of aPL on trophoblasts is neutrophil activation. Indeed, pregnant mice infused with aPL show neutrophil infiltration in the placenta and the deleterious effects of aPL on fetal survival and growth are significantly reduced by neutrophil depletion [120]. Furthermore, neutrophils may also be directly activated by anti-β2-GPI antibodies that recognize β2-GPI bound to their cell surface and stimulate NETs formation through mechanisms dependent on ROS and TLR-4 [127]. Increasing evidence demonstrates that neutrophils are related to obstetric APS, in which pathogenic NETosis is initiated by aPL binding to trophoblasts. In individuals with SLE and with preeclampsia, an increased number of NETs were found infiltrating placental intervillous spaces, in association with marked inflammatory and vascular modifications [128].

In clinical practice there are women with a history of recurrent early abortions or fetal loss, but persistently negative for conventional aPL laboratory tests and thus classified as obstetrical SN-APS (OSN-APS). Since APS is the most common treatable cause of recurrent miscarriage, it is important to provide an appropriate diagnosis to help treat cases of women with OAPS and prevent pregnancy from failing. Regarding this specific subpopulation of pregnant women, not fulfilling the diagnostic APS criteria, several studies describe the use of so-called non-criteria aPL tests (as reported in another section). In particular, a monocentric cohort study of OSN-APS highlights how non-criteria tests, such as immunostaining on thin layer chromatographic (TLC-immunostaining) and anti-vimentin/CL, were more sensitive than many other non-criteria biomarkers [129]. 

In obstetric APS, investigations on new tests capable of revealing undetectable positivity and the associated signal pathways, underlying a variety of unfavorable events in placental development, could be useful to the therapeutic strategy based on risk stratification [8,129]. Thus, it should include both the “aPL profile” and their molecular mechanisms as disease-specific factors.

## 6. Role of Antiphospholipid Antibodies in Seronegative APS

In daily clinical practice it may happen to find patients with a clinical profile suggestive of APS who are persistently negative on the routine aPL test. For these cases, the term seronegative APS has been proposed. Hughes himself, in describing this syndrome, proposes three possible arguments for SN-APS. First, that it is a misdiagnosis; secondly, that a previous positive aPL test has turned negative. Third, and most likely, the tests usually used for the detection of aPL are not enough, in some cases, or inadequate. The latter reason may depend on the limitations of traditional technical approaches or on the existence of antigenic targets other than those known. In recent years, new variants have emerged in aPL tests that support the no criteria aPL concept [19,130].

In this purpose, a different laboratory diagnostic approach was employed for the detection of aPL (anti-CL, anti-LBPA, anti-PS and anti-PE). TLC-immunostaining is a non-quantitative technique that identifies the reactivity of aPL in serum with purified phospholipid molecules. This test exploits the different binding characteristics of the phospholipid to the solid phase, which involves both electrostatic and hydrophobic interactions. Therefore, antigen exposure is quite different than that on the surface of the microtiter wells of the ELISA plate, where phospholipids are immobilized to coat the surface in single layer [131]. 

Recently, Zohoury et al. conclusively revealed a positive association between the presence of anti-PS/PT antibodies (IgG and/or IgM) and arterial or venous thrombosis. Binding of aPT antibodies on target molecules on the cell surface of endothelial cells was hypothesized as the pathogenetic mechanism of this aPL subtype [132]. aPT could be directed against both, PT and PS/PT-complexes. This same subgroup of aPL has been shown to be capable of inducing a procoagulant state through the activation of MAPK pathway [133]. To date, aPS/PT antibodies are not included in the APS laboratory criteria, but their positivity has been recently proposed as a part of the global APS score and has been shown to be a strong prognostic factor for both arterial and venous thrombosis. Litvinova E. et al. demonstrated the presence of IgG aPS/PT in 5.6% of SN-APS group and the presence of IgM in 16.7% of SN-APS group. Interestingly, a pathogenic role for IgG aPS/PT was suggested [134]. 

Moreover, Annexin A2 and Annexin A5 have been investigated. Anti-Annexin A5 antibodies have been proposed to be associated with the clinical features of APS, including thrombosis and recurrent miscarriages. β2-GPI dependent aPL may interfere with the protective binding of Annexin A5 to the endothelium, thus leading to thrombosis. Interestingly, resistance to Annexin A5 anticoagulant activity inversely correlates with titers of IgG antibodies targeting DI in both thrombotic and obstetric manifestations of SLE [135]. 

Annexin A2 is an important binding site for β2-GPI on the surface of ECs and monocytes and proved to be a component of a multimolecular signaling complex on the surface of ECs. Several studies have shown that cross-linking or clustering of annexin A2-bound β2-GPI leads to cellular activation resulting in expression of the procoagulant phenotype and inflammatory cytokines. Annexin A2 can be recognized by antibodies in serum from patients with systemic autoimmune disorders and anti-Annexin A2 antibodies were also detected in patients with APS [11]. Further studies are needed to determine their clinical significance and diagnostic value to discriminate against clinical subgroups of patients with APS [136]. Nevertheless, Annexin A2 has been identified as a target of autoantibodies in sera from patients with obstetric disorders, independently of others aPL and Annexin A5. In fact, the highest levels of anti-Annexin A2 were observed in sera from two patients with recurrent pregnancy loss and one patient with preeclampsia. These results suggested that anti-Annexin A2 antibodies may play a role in thrombotic mechanisms leading to placental integrity dysfunction [137]. To suggest the pathogenic role of these antibodies, it has been investigated the role of Annexin A2 in aPL pathogenicity in vitro, they studied the effect of an anti-Annexin A2 on aPL induced up-regulation of ICAM-1, E-selectin and TF on cultured ECs in vitro [138]. Moreover, a cell surface complex, Annexin A2 and its binding partner S100A10 (p11), was reported [139]. In addition, several data described the role of this complex in the generation of primary fibrinolytic protease and plasmin, highlighting how its regulation can be involved in hemostasis and thrombosis processes [139].

In the last few years, vimentin was shown to be able to bind CL in vitro, possibly as a result of electrostatic interaction between its positively charged amino acids and the negatively charged CL. Vimentin/cardiolipin complex was identified as a new target antigen of SN-APS, where vimentin is a new protein cofactor for CL. Serum IgG anti-vimentin/CL antibodies were found not only in a large proportion of SN-APS patients, but also in almost all APS patients. A pathogenic role of these autoantibodies was hypothesized, since affinity-purified anti-vimentin/CL antibodies from the sera of SN-APS patients were able to induce in vitro IRAK phosphorylation and NF-κB activation in endothelial cell [13,130]. Accordingly, IgG from SN-APS patients trigger the expression of VCAM-1, as well as release of TF from ECs, suggesting a biological activity for these antibodies with consequent proinflammatory and procoagulant effects. These data try to explain the paradoxical role of non-criteria aPL in seronegative APS, demonstrating how these unconventional antibodies, found in SN-APS patient sera, may contribute to pathogenesis of thrombosis and/or other clinical manifestations [13,129,130,140,141].

On the other hand, as mentioned above, autoantibodies against β2-GPI themselves are a heterogeneous population of antibodies containing subclasses directed against each β2-GPI domain [142].

Finally, several studies recently analyzed the role of the IgA isotype and identification of IgA aPL and/or anti-β2-GPI may represent an additional test in seronegative patients. Moreover, recent studies have suggested that while IgG/IgM isotypes recognize an epitope in β2-GPI domain I, the epitopes recognized by IgA are in domains III, IV and V, showing the role of these antibodies as pathogenetic [135]. Indeed, some authors, using mouse model of thrombosis, have demonstrated the pathogenicity of purified IgA anti-β2-GPI antibodies specific for epitopes of β2-GPI domains IV/V, highlighting that these antibodies may be found in association with thrombosis, especially arterial thrombosis [62]. The pathogenicity of IgA aPL was also demonstrated in another work, where results proved that mice, injected with IgA aPL from patients with APS, developed thrombosis [143]. IgA isotype of the aCL is detectable in 4% of SN-APS patients, and IgA anti-β2-GPI antibodies only in 2%. Only few studies showed that IgA anti-β2-GPI antibodies were significantly increased in women with pregnancy morbidity [129]. Nevertheless, various studies described the association between the exclusive detection of IgA anti-β2-GPI antibodies and clinical manifestations of APS. For instance, women with unexplained recurrent spontaneous abortions and fetal death were positive for IgA isotype of anti-β2-GPI antibodies and negative for LA [142]. In the same way, different retrospective and prospective studies recommend to consider the presence of IgA anti-β2-GPI antibodies as a further laboratory criteria, to increase the number of APS patients diagnosed [144]. However, as reported by the 13th International Congress on Antiphospholipid Antibodies, the test for IgA-anti-β2-GPI should be considered useful in patients negative for IgG and IgM isotypes with APS symptoms, thus could be considered as a non-criteria test to detect aPL in SN-APS patients [145] (Table 1).

Therefore, aPL proves to be an extremely heterogeneous family of antibodies; in fact, more than 30 different antibodies have been described in patients with APS, so much so that it deserves the nickname of “explosion of autoantibodies in APS” [146]. Recent evidence strongly suggests that also in sera from patients with SN-APS it is possible to detect a large spectrum of antibodies, by using new antigenic targets or methodological approaches, and that these antibodies may play a functional role in the pathogenesis of clinical manifestations (Figure 3).

## 7. Conclusions

The pathogenetic mechanisms of the so-called seronegative syndrome contain a contradiction of terms. Indeed, as long as the pathology remains seronegative, we do not have molecular referents to attribute damage effector mechanisms. When we are able to reveal the molecules associated with the disease, the autoantibodies in this case, we can no longer speak about seronegative APS.

In this case, the laboratory diagnosis of SN-APS is related to precise recognized criteria which, in the specific case, are not fulfilled and, therefore, it is still referred to as seronegative.

Conversely, in this review we focus on the possibility of framing these particular cases of the syndrome with a laboratory diagnosis that relies on what are recognized as non-criteria, and which prompt us to reveal antibodies and identify, or at least hypothesize, pathogenetic mechanism(s).

The various aspects addressed in this review suggest pathogenetic pathways that are mostly attributable to molecular mechanisms that could also play a role in SN-APS. In particular, the signal transduction pathway triggered by TLR-4 within lipid rafts, with involvement of MyD88 and IRAK, NF-κB activation and consequent release of proinflammatory and procoagulant factors has been elucidated.

Searching not only for specific antibodies, but also investigating their functional role, i.e., their capability to trigger a signal transduction pathway(s), may represent the new approach for identifying true pathogenic aPL. It may represent the starting point of clinical management and therapeutic treatment of seronegative APS.

## Figures and Tables

**Figure 1 ijms-21-08411-f001:**
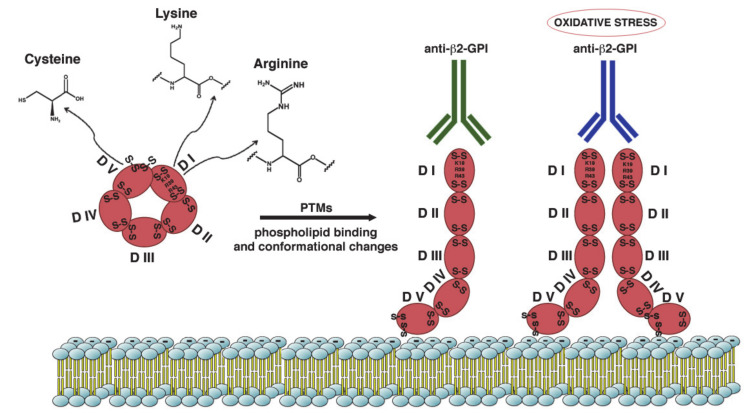
**Schematic representation of β2-GPI structure and conformation.** β2-GPI consist in five domains (I-V) with two disulfide bonds in each domain and an additional disulfide bond in domain V. Phospholipid binding by domain V and some PTMs of the protein result in a conformational change from the circular (closed) form to open configuration. This unfolded conformation may facilitate the exposition of ‘‘cryptic epitope’’ and autoantibodies binding. The amino acids most involved in the PTMs are Lysine, Arginine and Cysteine. Increased oxidative stress may alter the configuration of β2-GPI to a dimeric form that enhance antibody affinity.

**Figure 2 ijms-21-08411-f002:**
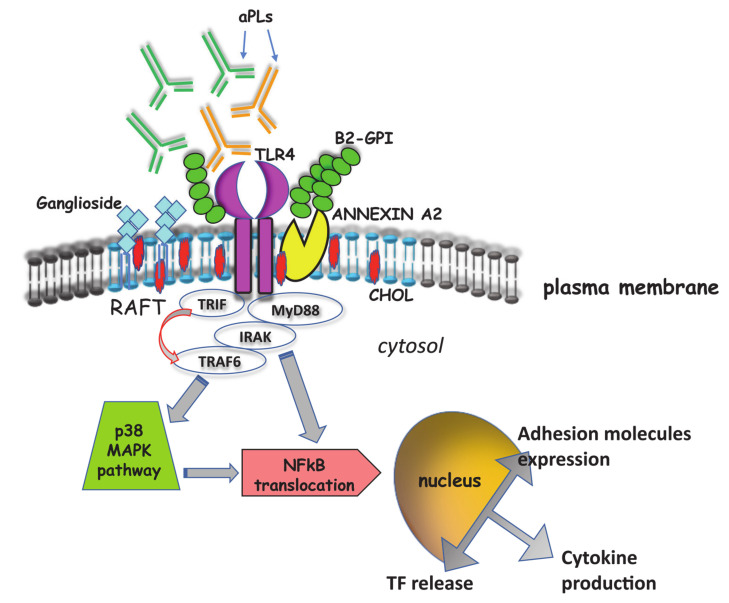
**Signaling pathway induced by aPL.** Schematic drawing depicting the signaling transduction pathways triggered by aPL via TLR-4, through lipid rafts. This pathway leads to NF-κB activation and translocation to the nucleus, with an increase of proinflammatory cytokines and chemokines production, adhesion molecules expression and tissue factor release.

**Figure 3 ijms-21-08411-f003:**
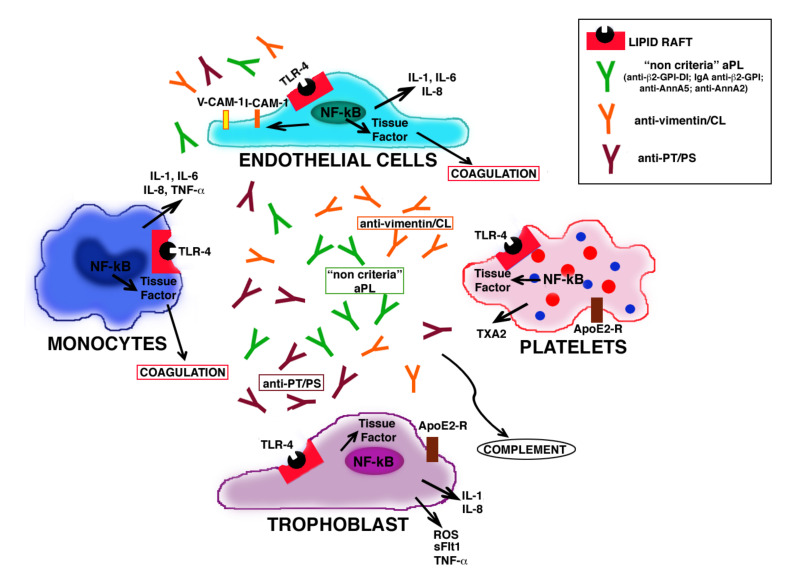
**Putative pathogenetic mechanism of SN-APS.** Molecular mechanisms of non-criteria aPL may involve endothelial cells and trophoblast cells with monocyte and platelet cooperation, leading to a proinflammatory and procoagulant state that underlies thrombosis and pregnancy complications.

**Table 1 ijms-21-08411-t001:** Different non-criteria specificities in seronegative Antiphospholipid Syndrome (SN-APS).

Non-Criteria aPL	Clinical Association	References
Anti-β2-GPI-DI	association with thrombosis and pregnancy complications (more than antibodies directed to other domains)	[53,54,55,56,57,58,59]
IgA anti-β2-GPI	correlation with thrombosis, miscarriages, pulmonary hypertension, seizure, thrombocitopenia and livedo reticularis	[62,143,145]
Anti-PT/PS	strongly correlation with thrombosis and obstetric manifestations; in association with other anti-PL contribute to assess the risk of thrombosis	[132,133,134,135]
TLC-immunostaining detection of anti-CL	TLC-immunostaining could potentially identify the presence of aPL in SN-APS; this is in association with vascular thrombosis	[130,140,141]
Anti-Vimentin/CL	correlation with thrombotic events and pregnancy morbidity	[13]
Anti-AnnA5	clinical correlation with pregnancy-related morbidity is still controversial	[135]
Anti-AnnA2	alter pro-fibrinolytic activity; correlate with thrombotic events	[136,137,138]

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
