# Peer review of "Molecular Mechanisms of “Antiphospholipid Antibodies” and Their Paradoxical Role in the Pathogenesis of “Seronegative APS”"

_ijms, 2020, doi:10.3390/ijms21218411_

Round 1

Reviewer 1 Report

In this interesting study authors describes the paradoxical role of antiphospholipid antibodies (aPL)in the pathogenesis of "seronegative APS".  Most of the information contained in it is of general utilityand graphics / schemes are appropriate.

However there are some aspects that should be improved.

The reading is difficult for an average reader who does not dedicate himself exclusively or preferentially to APS. The first problema is the dispersion of the information, the lack of explanation of key concepts in the appropriate moments. So the manuscript should be restructured to go from the most general to the most specific information.

Abstract
The abstract is confusing and unstructured. Although the title of the paper names the seronegative APS, the abstract does not introduce the concept of APS or the concept of seronegative APS. In a review article, the abstract should be more general and mention only the essential issues to understand what the article is about.

It should be recomposed to facilitate easy comprehension by readers who are not fully integrated into the APS (as we will also comment in other parts of this review-report).

 The abstract has to serve to captivate the reader's attention and thus encourage him to read the entire article. Keep in mind that this article, as it is a review, will be read by scientists with different levels of knowledge of the APS

The introduction should clearly explain the concept of APS as well as the classification criteria of APS (both clinical and laboratory) from the first lines of the manuscript.
In addition to distinguishing between pathogenic and non-pathogenic PLA, it is essential to make a distinction between criteria-aPL (those agreed at the 2004 Sidney meeting, Miyakis et al, reference 30 of the manuscript) and the extra-criteria aPL, those not included in the consensus Sidney. Somo of the exyra-criteria were listed in the publication from Miyakis et al.

Likewise, the concept of APS and seronegative APS should be clear from the first moments, a few paragraphs before the place where they are in the manuscript (from line 88), so that the reader with less experience in APS can progressively assimilate the information.

It would be interesting describe the situations that can lead to a seronegative APS, described by Hughes and Khamashta in 2003 (diagnostic error, aPL negativization, problems with the diagnostic systems or presence of extra- aPL criteria). By the way this article should be cited immediately after the 2003 number on line 88 (PMID: 14644846).

Line 36: The authors indicate that there are two types of antigenic targets for aPLs. However, these antibodies can recognize phospholipids, phospholipid-binding proteins or antigens formed by the combination of both types of molecules; in fact, anti-cardiolipin recognize the Cardiolipin / B2GP1 complex. Please reorganize this paragraph because it remains somewhat complex for the reader with moderate experience in APS.

Once these three types of targets have been explained, you can go on to explain which specific proteins and phospholipids can be targeted by aPL.
For ease of reading, it is best to describe the antigens that correspond to the three types of targets in three separate sentences.
Note that the paragraph has 12 references. It would be more useful to place these references in the specific place where appropriate, and not all together.

When referring to pathogenic or non-pathogenic antibodies(Line 47): it should be explained previously that the presence of aPLs by itself is not enough to trigger the APS event. In this sense, all aPLs are "non-pathogenic" during the time that patients are asymptomatic carriers. This would be the right place to discuss the two hits  hypothesis (treated in Line 100 and following). 

Line 105, Is a mistake ?: Protein C is Vitamin K-dependent, not Vitamin C (ascorbic acid)

The information about the various conformations of B2GP1, with different antigen-exposing abilities should be explained in advance. For example immediately after describing the 5 domains. The information on the peculiarities of the different structural forms can be left where they are.

Paragraph that ends on line 180. This paragraph should be rewritten. Although domain 1 is of great importance for the pathogenesis of APS, pathogenic aPLs targeting all 5 domains of B2GP1 have been described (PMID: 12905463). In the 16th International Congress on Antiphospholipid Antibodies several communications (P23, P45, p46) dealt aPL against domains other than D1   (DOI: 10.1177/0961203319872604)-

Lines 196-197.    The afirmation  "On the contrary, antibodies targeting the fourth and the fifth domain (DIV/V) of β2-GPI are not directly related to thrombosis" must be modified. It has been described pathogenicity associated with aPL directed against this zone of B2GP1.

The group of Pierangeli demonstrated that anti-β2 GPI antibodies directed to domain IV/V of β2 GPI represent an important subgroup of clinically relevant antiphospholipids and demonstrated their patogenicity  in a mouse model of thrombosis.  (PMID: 23983008).

In the same way,  Blank et al. (Shoenfeld’ group) identified several  human monoclonal antibodies anti-B2GPI (from APS patients)  that  recognized antigens in domains 3 and 4 of B2GP1 and induce experimental thrombotic APS in mice models.  In adition, ono of this, the monoclonal ILA-3, induce also gestational-APS in mice models (PMID: 9521339). ILA-3   recognized the peptide 208-KDKATF-213 (fourth domain of the B2GPI). The monoclonal H3 recognize Peptide 133-TLRVYK-138 (this corresponds to the third domain) (PMID: 2784733). Importance of APS-associated antibodies directed to ILA-3 antigen in domain 4) has been described recently by other groups: Serrano el al.(PMID: 31134087) and Sippl et al (communicated in 2019 in the 16th International Antiphospholipid antibodies meeting), this Swedish  group has not published the results, it has only communicated them in meetings (are under patent process).

The fact that there are authors who do not find a clinical association with the presence of anti Domain 4 antibodies may almost certainly be due to the use of recombinant D4/5.   The commercially availables  B2GP1-D4/5 recombinant proteins are truncated forms and lack the first 33 amino acids of domain 4. The antigenic zone of domain 4 is located in the first amino acids and therefore these recombinant proteins do not contain the epitope ILA-3 (PMID: 31134087).

Although the review deals with anti-phospholipid antibodies in Seronegative APS, the space used for the two most important extra-criteria aPLs: anti-phostadidylserine / prothrombin and, especially, anti-B2GP1 of IgA isotype is very limited.  
There are many articles about the association of the presence of isolate IgA anti B2GP1 (negative for other aPLs) with thrombosis. So, it has been clearly demonstrated this association, both in retrospective studies (PMID: 23983008, PMID: 24741618) and prospective studies (PMD: 23538147, PMID: 22358146).

In Table 1, among the APS events associated with anti-B2GP1 IgA, surprisingly thrombosis is not mentioned and no references are provided about thrombosis and IgA isotype. Please correct it.
The long-term risk of thrombosis in asymptomatic carriers of  isolate IgA anti B2GP1 (negative for other aPL) has also been. IgA anti B2GP1 carriers have a rate of APS events / year similar to that of IgG anti B2GPI carriers (PMID: 28727732)

It is also true that there are articles in which no association was found between the presence of anti B2GP1 IgA and thrombosis. This situation is originated because for IgA  isotype of aPL there are no standardization levels that other aPLs (PMID: 24245938). So many authors who use inadequate detection systems consider their patients as negative (false negatives). These controversies and the lack of standardization for the extra-criteria aPL should be addressed in the review.

The treatment of anti Anexin A2 is also too limited. It should be expanded regarding autoantibodies against the molecules of the complex that regulates the generation of plasmin (Anti Annexin A2 and Anti S100A10). The decrease in fibrinolysis produced by these blocking antibodies is also a cause of SN-APS  thrombosis. (PMID: 18322784, PMID: 23193360, PMID: 27631133, PMID: 31059997, PMID: 32115196, PMID: 27372915).
Citation 80, about the association of annexin A2 with B2GP1 is very old. There are more recent publications. PMID: 27449504, PMID: 15471954, PMID: 18827060, PMID: 25533130 …

The references cited in the evaluation have been included to document the review and, where appropriate, to guide how to improve specific aspects.

The authors are free to use these references or other similar ones that they consider more suitable for later editions of the manuscript.

Author Response

Referee 1

In this interesting study authors describes the paradoxical role of antiphospholipid antibodies (aPL)in the pathogenesis of "seronegative APS".  Most of the information contained in it is of general utility and graphics / schemes are appropriate.

However there are some aspects that should be improved.

The reading is difficult for an average reader who does not dedicate himself exclusively or preferentially to APS. The first problema is the dispersion of the information, the lack of explanation of key concepts in the appropriate moments. So the manuscript should be restructured to go from the most general to the most specific information.

We basically agree with the criticisms highlighted by the Referee, we have therefore modified what is suggested as follows: 

Abstract
The abstract is confusing and unstructured. Although the title of the paper names the seronegative APS, the abstract does not introduce the concept of APS or the concept of seronegative APS. In a review article, the abstract should be more general and mention only the essential issues to understand what the article is about.

It should be recomposed to facilitate easy comprehension by readers who are not fully integrated into the APS (as we will also comment in other parts of this review-report).

 The abstract has to serve to captivate the reader's attention and thus encourage him to read the entire article. Keep in mind that this article, as it is a review, will be read by scientists with different levels of knowledge of the APS

The Abstract has been completely re-written, according to the wise suggestions.

The introduction should clearly explain the concept of APS as well as the classification criteria of APS (both clinical and laboratory) from the first lines of the manuscript.
In addition to distinguishing between pathogenic and non-pathogenic PLA, it is essential to make a distinction between criteria-aPL (those agreed at the 2004 Sidney meeting, Miyakis et al, reference 30 of the manuscript) and the extra-criteria aPL, those not included in the consensus Sidney. Somo of the exyra-criteria were listed in the publication from Miyakis et al.

Likewise, the concept of APS and seronegative APS should be clear from the first moments, a few paragraphs before the place where they are in the manuscript (from line 88), so that the reader with less experience in APS can progressively assimilate the information.

It would be interesting describe the situations that can lead to a seronegative APS, described by Hughes and Khamashta in 2003 (diagnostic error, aPL negativization, problems with the diagnostic systems or presence of extra- aPL criteria). By the way this article should be cited immediately after the 2003 number on line 88 (PMID: 14644846).

Line 36: The authors indicate that there are two types of antigenic targets for aPLs. However, these antibodies can recognize phospholipids, phospholipid-binding proteins or antigens formed by the combination of both types of molecules; in fact, anti-cardiolipin recognize the Cardiolipin / B2GP1 complex. Please reorganize this paragraph because it remains somewhat complex for the reader with moderate experience in APS.

Once these three types of targets have been explained, you can go on to explain which specific proteins and phospholipids can be targeted by aPL.
For ease of reading, it is best to describe the antigens that correspond to the three types of targets in three separate sentences.
Note that the paragraph has 12 references. It would be more useful to place these references in the specific place where appropriate, and not all together.

When referring to pathogenic or non-pathogenic antibodies (Line 47): it should be explained previously that the presence of aPLs by itself is not enough to trigger the APS event. In this sense, all aPLs are "non-pathogenic" during the time that patients are asymptomatic carriers. This would be the right place to discuss the two hits hypothesis (treated in Line 100 and following). 

We really thank the Referee for being so precise and thorough in the comments. We agree with the criticisms raised and we have followed the directions, modifying the drafting of the Introduction section, in particular, we better define the APS and the classification criteria from the first lines of our ms.  The SN APS concept follow immediately, describing the situations that can lead to SN APS, with the right references (ref. 19 and 20).

Moreover, we described the aPL antigens in three separate sentences, with specific references.

Line 105, Is a mistake?: Protein C is Vitamin K-dependent, not Vitamin C (ascorbic acid)

Yes! Of course. We corrected it.

The information about the various conformations of B2GP1, with different antigen-exposing abilities should be explained in advance. For example immediately after describing the 5 domains. The information on the peculiarities of the different structural forms can be left where they are.

Paragraph that ends on line 180. This paragraph should be rewritten. Although domain 1 is of great importance for the pathogenesis of APS, pathogenic aPLs targeting all 5 domains of B2GP1 have been described (PMID: 12905463). In the 16th International Congress on Antiphospholipid Antibodies several communications (P23, P45, p46) dealt aPL against domains other than D1   (DOI: 10.1177/0961203319872604)-

Lines 196-197.    The afirmation  "On the contrary, antibodies targeting the fourth and the fifth domain (DIV/V) of β2-GPI are not directly related to thrombosis" must be modified. It has been described pathogenicity associated with aPL directed against this zone of B2GP1.

The group of Pierangeli demonstrated that anti-β2 GPI antibodies directed to domain IV/V of β2 GPI represent an important subgroup of clinically relevant antiphospholipids and demonstrated their patogenicity  in a mouse model of thrombosis.  (PMID: 23983008).

In the same way, Blank et al. (Shoenfeld’ group) identified several  human monoclonal antibodies anti-B2GPI (from APS patients)  that  recognized antigens in domains 3 and 4 of B2GP1 and induce experimental thrombotic APS in mice models.  In adition, ono of this, the monoclonal ILA-3, induce also gestational-APS in mice models (PMID: 9521339). ILA-3   recognized the peptide 208-KDKATF-213 (fourth domain of the B2GPI). The monoclonal H3 recognize Peptide 133-TLRVYK-138 (this corresponds to the third domain) (PMID: 2784733). Importance of APS-associated antibodies directed to ILA-3 antigen in domain 4) has been described recently by other groups: Serrano el al.(PMID: 31134087) and Sippl et al (communicated in 2019 in the 16th International Antiphospholipid antibodies meeting), this Swedish  group has not published the results, it has only communicated them in meetings (are under patent process).

The fact that there are authors who do not find a clinical association with the presence of anti Domain 4 antibodies may almost certainly be due to the use of recombinant D4/5.   The commercially availables  B2GP1-D4/5 recombinant proteins are truncated forms and lack the first 33 amino acids of domain 4. The antigenic zone of domain 4 is located in the first amino acids and therefore these recombinant proteins do not contain the epitope ILA-3 (PMID: 31134087).

Following Referee’s suggestions, we reorganized the section and reported more in details the pathogenicity of different Domains. We added specific and pertinent references.

Although the review deals with anti-phospholipid antibodies in Seronegative APS, the space used for the two most important extra-criteria aPLs: anti-phostadidylserine / prothrombin and, especially, anti-B2GP1 of IgA isotype is very limited.  
There are many articles about the association of the presence of isolate IgA anti B2GP1 (negative for other aPLs) with thrombosis. So, it has been clearly demonstrated this association, both in retrospective studies (PMID: 23983008, PMID: 24741618) and prospective studies (PMD: 23538147, PMID: 22358146).

In Table 1, among the APS events associated with anti-B2GP1 IgA, surprisingly thrombosis is not mentioned and no references are provided about thrombosis and IgA isotype. Please correct it. We did it!
The long-term risk of thrombosis in asymptomatic carriers of isolate IgA anti B2GP1 (negative for other aPL) has also been. IgA anti B2GP1 carriers have a rate of APS events / year similar to that of IgG anti B2GPI carriers (PMID: 28727732)

It is also true that there are articles in which no association was found between the presence of anti B2GP1 IgA and thrombosis. This situation is originated because for IgA  isotype of aPL there are no standardization levels that other aPLs (PMID: 24245938). So many authors who use inadequate detection systems consider their patients as negative (false negatives). These controversies and the lack of standardization for the extra-criteria aPL should be addressed in the review.

The treatment of anti Anexin A2 is also too limited. It should be expanded regarding autoantibodies against the molecules of the complex that regulates the generation of plasmin (Anti Annexin A2 and Anti S100A10). The decrease in fibrinolysis produced by these blocking antibodies is also a cause of SN-APS  thrombosis. (PMID: 18322784, PMID: 23193360, PMID: 27631133, PMID: 31059997, PMID: 32115196, PMID: 27372915).
Citation 80, about the association of annexin A2 with B2GP1 is very old. There are more recent publications. PMID: 27449504, PMID: 15471954, PMID: 18827060, PMID: 25533130 …

 As suggested, we added more reported data on extra-criteria aPL with specific citations.

We updated the citation on Annexin A2/beta2 with a more recent publication (new ref. 88)

The references cited in the evaluation have been included to document the review and, where appropriate, to guide how to improve specific aspects.

The authors are free to use these references or other similar ones that they consider more suitable for later editions of the manuscript.

Reviewer 2 Report

  It is an interesting review article to discuss the molecular mechanisms of antiphospholipid antibodies (APL) and antibodies against oxidized/glycated b2-GPI in antiphospholipid syndrome (APS), and non-criteria APL in the pathogenesis of seronegative APS. The mechanisms include Toll-like receptor (TLR) induction, intracellular activation of MyD88-dependent and -independent signaling, leading to activation of NF-κB and production of cytokines, adhesion molecules and tissue factor as well as activation of complements.

  The manuscript is well written in English and the content are relevant to the clinical application. There is no further comments on the manuscript except the content of Fig. 2. In the section 4. Signal transduction pathways triggered by APL, the authors mentioned that engagement of TLR-4 activates MyD88-dependent and -independent pathways. However, only MyD88-dependent pathway was shown in Fig. 2. According to Trends in Biochemical Sciences 2012;37:92, there is an MyD88-independent signaling via TRIF to activate downstream NF-κB, and the TRIF can also trigger the MyD88-dependent TRAF6, leading to downstream activation of NF-κB, JN K and p38 MAPK.

Author Response

Referee 2

It is an interesting review article to discuss the molecular mechanisms of antiphospholipid antibodies (APL) and antibodies against oxidized/glycated b2-GPI in antiphospholipid syndrome (APS), and non-criteria APL in the pathogenesis of seronegative APS. The mechanisms include Toll-like receptor (TLR) induction, intracellular activation of MyD88-dependent and -independent signaling, leading to activation of NF-κB and production of cytokines, adhesion molecules and tissue factor as well as activation of complements.

  The manuscript is well written in English and the content are relevant to the clinical application. There is no further comments on the manuscript except the content of Fig. 2. In the section 4. Signal transduction pathways triggered by APL, the authors mentioned that engagement of TLR-4 activates MyD88-dependent and -independent pathways. However, only MyD88-dependent pathway was shown in Fig. 2. According to Trends in Biochemical Sciences 2012;37:92, there is an MyD88-independent signaling via TRIF to activate downstream NF-κB, and the TRIF can also trigger the MyD88-dependent TRAF6, leading to downstream activation of NF-κB, JN K and p38 MAPK.

Thank you for the positive comments.

We modified Fig.2, which includes the new MyD88-independent signaling via TRIF.

Round 2

Reviewer 1 Report

The questions and suggestions planted have been resolved correctly